# Connection of Conic and Cylindrical Map Projections

**Miljenko Lapaine** 

Faculty of Geodesy, University of Zagreb, Kačićeva 26, 10000 Zagreb, Croatia; mlapaine@geof.hr

**Abstract:** In previous papers that have dealt with cylindrical map projections as limiting cases of conical projections, standard or equidistant parallels were used in the derivations. This paper shows that this is not necessary and that it is sufficient to use parallels that preserve length. In addition, unlike other approaches, in this article the limiting cases of conic projections are derived in the most natural way, by deriving the equations of cylindrical projections from the equations of conic projections in a rectangular system in the projection plane using a mathematical concept of limits. It is shown that such an approach is possible, but not always, so it should be used carefully, or even better, avoided in teaching and studying map projections.

**Keywords:** conic map projection; cylindrical map projection; length preservation; limit

## 1. Introduction

Monmonier [1] (pp. 75–76) wrote:

"Lambert's contributions to cartography include seven different map projections as well as an illuminating mathematical analysis of conformality. In addition to using calculus to derive Bond's analytical formula for the Mercator projection, he demonstrated that the Mercator map is a "special case" in a family of conformal projections with polar and conic versions. . . . the cylinder and the plane are extreme forms of a cone tangent to the sphere along a "standard parallel". Positioning the apex at infinity converts the cone to a cylinder, with the standard parallel at the equator. Putting the apex on the North Pole flattens the cone to a plane and shrinks the standard parallel (at 90°) to a point. If the projections are conformal, the cylindrical case is the Mercator, the planar case is the polar stereographic (in use since about 150 BC), and all intermediate cases are instances of the Lambert conformal conic projection, presented in 1772."

At first glance, in the mentioned quote, everything looks fine, but Lambert does not perceive conic and cylindrical projections as projections on the surfaces of a cone or cylinder, which then develop into a plane. In contrast, Lambert [2] mentions only at the end of his exposition in §56 that a map made in a conformal conic projection can be folded into a cone! Thus, Monmonier (2004) unfairly attributes to Lambert something that is not true.

Apart from Monmonier [1], several other authors mention that cylindrical and azimuthal projections can be interpreted as limiting cases of conic projections [3–11]. However, there are few attempts to prove this claim [2,12–16].

Lapaine [15] set himself a goal to show in a rigorous and systematic way how to generally approach solving the problem of transition from a conic to a corresponding cylindrical projection and vice versa. First, he briefly explained the shortcomings of the previously known derivations. Then, he interpreted and supplemented Lambert's derivation, which leads from a conic conformal projection to a cylindrical conformal projection, i.e., the Mercator projection. Following Lambert's idea, Lapaine showed that not only conformal, but also equivalent and equidistant cylindrical projections can be derived from conic map projections. Lapaine also showed that the transition from the conic to the corresponding cylindrical projection is not always possible. Lapaine found the connection between conic and cylindrical projections in the differential equations that define these projections.

Lapaine [16] addresses the same problem, but this time supplements the explanations and derivations of Hinks [12], who derives a cylindrical equidistant projection from a conic projection equidistant along the meridians (simple conic). From the conformal conic projection, he derives the conformal cylindrical projection, that is, the normal aspect Mercator projection. For a simple equivalent projection with one standard parallel, Hinks gives no derivation. Lapaine [16] gives a derivation for the equivalent cylindrical projection as a limiting case of the equivalent conic projection and a derivation for the central perspective cylindrical projection as a limiting case of the central conic perspective projection. Lapaine assumed that an equidistantly mapped parallel in the conic projection would be mapped to an equidistantly mapped parallel in the corresponding cylindrical projection.

In regard to the interpretation of cylindrical and azimuthal projections as limiting cases of conic projections, auxiliary developable surfaces are usually used (Figure 1). If the angle at the top of the cone becomes larger and becomes right at one point, the cone becomes a plane. The conclusion follows from there: if conic projections are mappings onto a cone, then the limiting case of these projections should be azimuthal projections. If the angle at the top of the cone decreases to zero at one point, the cone moves in a straight line, but if at the same time one of its cross sections maintains its size, the limiting case is a cylindrical surface. And now the conclusion naturally arises that if conic projections are mappings onto a cone, then the limiting case of these projections should be cylindrical projections. In this paper, it is shown that this is not always the case, offering another proof of the claim that the use of intermediate surfaces in the theory of map projections is generally not recommended. Intuitiveness does not always lead to the correct conclusion.

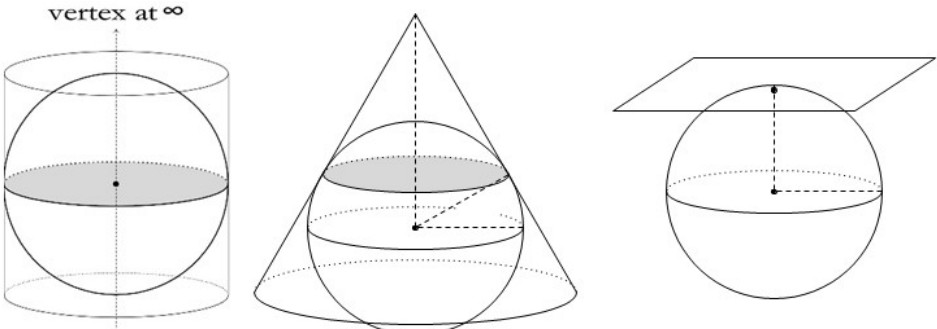

**Figure 1.** Cone in the middle and limiting cases that should correspond to the Mercator (**left**) and stereographic projection (**right**), according to [14].

Daners [14] derives the Mercator projection equations as the limiting case of a conformal conic projection of a sphere onto a cone that touches the sphere along one parallel (Figure 1). Daners does not deal with equidistant, equivalent, or other projections.

It should be noted that Daners [14], as well as Hoschek [13], but also many other authors experience the following (Figure 1):

(a) Cylindrical projections as projections onto the cylindrical surface which, after projection, develops into a plane;

(b) Conic projections as projections onto the conic surface which, after projection, develops into a plane;

(c) Azimuthal projections as projections directly onto a plane that touches the sphere at one point.

Such an approach is not good in principle because the vast majority of projections are direct mappings of a sphere or ellipsoid into a plane.

Cartographic projections are mappings of a curved surface, usually a sphere or an ellipsoid, onto a plane. The introduction of development surfaces as intermediate surfaces and the interpretation according to which a sphere or ellipsoid is first mapped onto such a surface and then developed into a plane is a fabrication that mostly does not correspond

to reality. It is usually justified by the term "conceptual" and explained by the claim that cartographic projections are easier to interpret that way.

The classification of map projections according to developable surfaces is useful for understanding the selection of projections and their parameters. However, although development surfaces are a useful conceptual tool, it should be emphasized that most map projections cannot be constructed geometrically but are instead defined mathematically.

Cartographic projections are usually classified according to the geometric surface from which they are derived: cylinder, cone, or plane. However, such an approach is correct only at first glance. It is actually the opposite. Cylindrical projections are not so called because they are mapping onto a cylindrical surface, but because the map made in such a projection can be folded into a cylinder. Similarly, conic projections are not so called because they map onto a conical surface, but because a map made in a conic projection can be folded into a cone.

Authors who today describe map projections in great detail using developable surfaces may not even be aware that in this way they are introducing double mappings into the theory of map projections. First, the Earth's sphere is mapped onto an auxiliary developable surface, and then it is transformed in some way, for example by development, into a plane map.

Double mappings have their role in the theory of cartographic projections in some special cases, but not in general. It should be known that developing into a plane is an isometry, i.e., such a mapping that preserves distances.

The use of developable surfaces in the definition of a cartographic projection is justified only for a small number of projections. Namely, in addition to cylindrical and conical projections, there are many others, such as azimuthal, pseudocylindrical, pseudoconical, conditional, etc., which cannot be interpreted by mapping onto the surface of a cylinder or cone.

There are some attempts in this direction. For example, a pseudocylindrical projection is said to be a mapping to a pseudocylinder, but it does not say what a pseudocylinder is. Nor is such a projection interpreted as a mapping onto an oval surface, without noticing that it is not a surface that can be developed.

In this article, we derive the equations of cylindrical equidistant, equivalent, conformal, and perspective projections from the equations of the corresponding conic projections in a mathematically correct way. In the end, we conclude that such an approach does not always give the expected result, and therefore its use is not recommended.

## 2. From Conic to Cylindrical Map Projections

The equations of normal aspect conic projections are usually given in the polar coordinate system:

$$\theta = m\lambda, \ \rho = \rho(\varphi), \tag{1}$$

where $\varphi$ and $\lambda$ are geographical coordinates, $\varphi \epsilon \left[ -\frac{\pi}{2}, \frac{\pi}{2} \right]$, $\lambda \epsilon [-\pi, \pi]$, $m$ the parameter, $0 < m < 1$, and $\theta$ and $\rho$ are polar coordinates in the projection plane. The function $\rho = \rho(\varphi)$ should be continuous, differentiable, and monotonic.

It is often explained that for $m = 1$, the equations of azimuthal projection are obtained from (1), and for $m = 0$, the equations of cylindrical projection are obtained. For the choice of $m = 1$, it is obvious that we obtain an azimuthal projection with the property $\theta = \lambda$, which by definition gives an azimuthal projection. However, for the choice of $m = 0$, we obtain $\theta = 0$, which does not give any map projection.

Therefore, to obtain a cylindrical projection from (1) with $m = 0$, we must add another condition to prevent the image of the sphere from being compressed into a straight line. In the following sections, we will demonstrate the possibility of obtaining cylindrical projections as limiting cases of conic projections using examples of equidistant, equivalent, conformal, and perspective projections.

Let us see what happens to Equation (1) when $m$ approaches zero, i.e., let us investigate the limiting case $m \to 0$. Unlike the derivations in the articles by [15,16], in this paper

we do not start with the mapping given in the polar coordinate system but rather in the rectangular coordinate system in the plane.

In addition, we do not assume the existence of a standard or equidistantly mapped parallel but rather start with a weakened assumption that there is a parallel that will be the true length mapped.

The equations of normal aspect conic projections in a rectangular coordinate (mathematical, right) system are as follows:

$$x = \rho(\varphi) \sin(m\lambda), \ y = \rho_0 - \rho(\varphi) \cos(m\lambda) \tag{2}$$

where $\rho_0 = \rho(\varphi_0)$ is the radius of a given parallel [17].

Let us first notice that

$$\lim_{m \to 0} \rho(\varphi) \sin(m\lambda) = \lim_{m \to 0} \rho(\varphi) \frac{\sin(m\lambda)}{m\lambda} m\lambda = \lim_{m \to 0} \rho(\varphi) m\lambda \tag{3}$$

because of

$$\lim_{m \to 0} \frac{\sin(m\lambda)}{m\lambda} = 1 \tag{4}$$

Furthermore,

$$\lim_{m \to 0} [\rho_0 - \rho(\varphi) \cos(m\lambda)] = \lim_{m \to 0} [\rho_0 - \rho(\varphi)], \tag{5}$$

because of

$$\lim_{m \to 0} \cos(m\lambda) = 1 \tag{6}$$

Let us mark

$$x = \lim_{m \to 0} \rho(\varphi) m\lambda, \ y = \lim_{m \to 0} [\rho_0 - \rho(\varphi)] \tag{7}$$

If there are limits (7) in the form of functions

$$x = x(\lambda), \ y = y(\varphi), \tag{8}$$

then these are the equations of the cylindrical projection.

For the existence of the limit $x = \lim_{m \to 0} \rho(\varphi) m\lambda$, it must be $\lim_{m \to 0} \rho(\varphi) = \infty$, and then for the existence of the limit $y = \lim_{m \to 0} [\rho_0 - \rho(\varphi)]$ it must be $\lim_{m \to 0} \rho_0 = \infty$. In that case, the first limit is of the form $\infty \cdot 0$, and the second $\infty - \infty$, which are indefinite expressions.

In principle, we can determine these limits on the assumption that there is a parallel that will be mapped in the true length by a conic projection. Let the parallel corresponding to the geographic latitude $\varphi_0$ be the parallel that preserves the true length when mapped onto a plane by conic projection (1). In the following, we will assume that we are mapping a sphere of radius 1. The length of its image in the projection plane will be equal to its length on the sphere

$$2\pi\rho_0 m = 2\pi \cos \varphi_0 \tag{9}$$

It follows from (9) that

$$m\rho_0 = \cos \varphi_0. \tag{10}$$

We will need this expression later. In addition, we will also need the factors of the local linear scale along the meridian, and the parallel. For conic projections of a sphere of radius 1, they read [8,17]:

$$h(\varphi) = -\frac{d\rho}{d\varphi}, \ k(\varphi) = \frac{m\rho}{\cos \varphi} \tag{11}$$

For cylindrical projections (8) and a sphere of radius 1, the local linear scale factors are [8,17]

$$h(\varphi) = \frac{dy}{d\varphi}, \ k(\varphi) = \frac{x}{\lambda \cos \varphi} \tag{12}$$

The problem of dividing by zero for $\varphi = \pm\frac{\pi}{2}$ appears, in other words the function $k = k(\varphi)$ in (11) and (12) is not defined for $\varphi = -\frac{\pi}{2}$ and $\varphi = \frac{\pi}{2}$.

In the following sections, we will explore cylindrical projections as limiting cases of some conic projections.

### 3. Projections Equidistant along the Meridians

For the normal aspect conic projection of the unit sphere given by (1) to be equidistant along the meridians, the condition

$$h = -\frac{d\rho}{d\varphi} = 1 \tag{13}$$

should be satisfied [8,17]. Solving the differential Equation (13) gives

$$\rho = K - \varphi, \tag{14}$$

where $K$ is a constant, $K \geq \frac{\pi}{2}$ to make $\rho \geq 0$ for each value of latitude. Therefore, in the polar coordinate system, the equations of the conic projection that is equidistant along the meridians read:

$$\theta = m\lambda, \ \rho = K - \varphi. \tag{15}$$

Considering (10), it is not difficult to obtain

$$x = \lim_{m \to 0} \rho(\varphi)m\lambda = \lim_{m \to 0} \frac{\rho(\varphi)}{\rho_0} m\rho_0\lambda = \lim_{K \to \infty} \frac{K - \varphi}{K - \varphi_0} \lambda \cos\varphi_0 = \lambda \cos\varphi_0, \tag{16}$$

$$y = \lim_{m \to 0}[\rho_0 - \rho(\varphi)] = \lim_{m \to 0}(\varphi - \varphi_0) = \varphi - \varphi_0 \tag{17}$$

Therefore, the equations of the normal aspect cylindrical projection equidistant along the meridians, which is the limiting case of the normal aspect conic projection equidistant along the meridians, for the unit sphere are

$$x = \lambda \cos\varphi_0, \ y = \varphi - \varphi_0. \tag{18}$$

At the same time, both projections give an image of one parallel $(\varphi_0)$ in the true length (9). If we translate the image of the projection by the amount $\varphi_0$ in the direction of the $y$ axis, we will achieve that the image of the equator is on the coordinate axis $x$ as is usual in the cartographic literature. Therefore, the final equations of the normal aspect cylindrical projection equidistant along the meridians, which is the limiting case of the normal aspect conic projection equidistant along the meridians, are

$$x = \lambda \cos\varphi_0, \ y = \varphi. \tag{19}$$

For the projection defined by Equation (19) we obtain

$$h(\varphi) = 1. \tag{20}$$

So, it is really a normal aspect cylindrical projection equidistant along the meridians.

*Example 1*

Let the conic projection be given by the equations

$$\theta = m\lambda, \ \rho = \frac{1}{2m} + \frac{\pi}{3} - \varphi, \tag{21}$$

where $m$ is a real number, $0 < m < 1$. The length of the parallel corresponding to the latitude $\frac{\pi}{3}$ will be on the unit sphere equal to $\cos\frac{\pi}{3} \cdot 2\pi = \pi$, and defined by (21) in the projection plane $\rho\left(\frac{\pi}{3}\right)m \cdot 2\pi = \frac{1}{2m}m \cdot 2\pi = \pi$.

The corresponding cylindrical projection has the following equations according to (19):

$$x = \frac{\lambda}{2}, \ y = \varphi, \tag{22}$$

and the length of any parallel in the projection (because they are all the same length) will be $2x(\pi) = 2\frac{\pi}{2} = \pi$. Therefore, the cylindrical projection (22) is the limiting case of the conic projection (21) when $m \to 0$, and both map the parallel of latitude $\frac{\pi}{3}$ so that its length in the plane of both projections is as long as that parallel on the unit sphere, i.e., $\pi$.

## 4. Equivalent Projections

For the normal aspect conic projection of the unit sphere given by (1) to be equivalent, the condition that the product of the factors of local linear scales along the meridians and along the parallels is equal to 1 must be satisfied, i.e., [8,17]

$$hk = -\frac{d\rho}{d\varphi}\frac{m\rho}{\cos\varphi} = 1. \tag{23}$$

Expression (23) is equivalent to

$$m\rho d\rho = -\cos\varphi d\varphi. \tag{24}$$

Integrating Equation (24) gives

$$\rho^2 = \frac{2}{m}(K - \sin\varphi), \tag{25}$$

where $K$ is a constant, $K \geq 1$ to make $\rho$ a real number for every value of latitude. Therefore, in the polar coordinate system, the equations of the conic equivalent projections read:

$$\theta = m\lambda, \ \rho = \sqrt{\frac{2}{m}(K - \sin\varphi)}. \tag{26}$$

Considering (10), we have

$$\rho_0 = \sqrt{\frac{2}{m}(K - \sin\varphi_0)} = \frac{\cos\varphi_0}{m}, \tag{27}$$

and from there

$$K = \frac{\cos^2\varphi_0}{2m} + \sin\varphi_0. \tag{28}$$

Furthermore,

$$x = \lim_{m \to 0}\rho(\varphi)m\lambda = \lim_{K \to \infty}\sqrt{\frac{K - \sin\varphi}{K - \sin\varphi_0}}\lambda\cos\varphi_0 = \lambda\cos\varphi_0, \tag{29}$$

$$y = \lim_{m \to 0}[\rho_0 - \rho(\varphi)] = \lim_{m \to 0}\frac{\rho_0^2 - \rho^2(\varphi)}{\rho_0 + \rho(\varphi)} = \lim_{m \to 0}\frac{\frac{2}{m}(\sin\varphi - \sin\varphi_0)}{\sqrt{\frac{2}{m}}\left[\sqrt{(K - \sin\varphi)} + \sqrt{(K - \sin\varphi_0)}\right]} =$$

$$= \lim_{m \to 0}\sqrt{\frac{2}{m}}\frac{\sin\varphi - \sin\varphi_0}{\sqrt{\frac{\cos^2\varphi_0}{2m} + \sin\varphi_0 - \sin\varphi} + \sqrt{\frac{\cos^2\varphi_0}{2m}}} =$$

$$= \lim_{m \to 0}2\frac{\sin\varphi - \sin\varphi_0}{\sqrt{\cos^2\varphi_0 + 2m(\sin\varphi_0 - \sin\varphi)} + \cos\varphi_0} = \frac{\sin\varphi - \sin\varphi_0}{\cos\varphi_0} \tag{30}$$

So, the equations of the normal aspect cylindrical equivalent projection of the unit sphere, which is the limiting case of the normal aspect conic equivalent projection, read:

$$x = \lambda \, \cos \varphi_0, \; y = \frac{\sin \varphi - \sin \varphi_0}{\cos \varphi_0}. \tag{31}$$

At the same time, both projections give an image of one parallel $(\varphi_0)$ in the true length (9). If we translate the image of the projection by the amount $\tan \varphi_0$ in the direction of the $y$ axis, we will achieve that the image of the equator is on the coordinate axis $x$ as is usual in the cartographic literature. Thus, the final equations of the normal aspect cylindrical equivalent projection, which is the limiting case of the normal aspect conic equivalent projection, are

$$x = \lambda \, \cos \varphi_0, \; y = \frac{\sin \varphi}{\cos \varphi_0}. \tag{32}$$

According to (12) for the projection (32) we have

$$h(\varphi) = \frac{\cos \varphi}{\cos \varphi_0}, \; k(\varphi) = \frac{\cos \varphi_0}{\cos \varphi} \tag{33}$$

So, it really is a normal aspect cylindrical equivalent projection.
*Example 2*
Let a conic projection be given by

$$\theta = m\lambda, \; \rho = \sqrt{\frac{1}{m}\left(\frac{1}{4m} + \sqrt{3} - 2\sin \varphi\right)}, \tag{34}$$

where $m$ is a real number, $0 < m < 1$. The length of the parallel corresponding to latitude $\frac{\pi}{3}$ will be on the unit sphere equal to $\cos \frac{\pi}{3} \cdot 2\pi = \pi$, and defined by (34) in the projection plane $\rho\left(\frac{\pi}{3}\right) m \cdot 2\pi = \frac{1}{2m} m \cdot 2\pi = \pi$.

The corresponding cylindrical projection has the following equations according to (32):

$$x = \frac{\lambda}{2}, \; y = 2\sin \varphi, \tag{35}$$

and the length of any parallel in the projection (because they are all the same length) will be $2x(\pi) = 2\frac{\pi}{2} = \pi$. Thus, the cylindrical projection (35) is the limiting case of the conic projection (34) when $m \to 0$, and both map the parallel of latitude $\frac{\pi}{3}$ so that its length in the plane of both projections is as long as that parallel on the unit sphere, i.e., $\pi$.

## 5. Conformal Projections

For the normal aspect conic projection of the unit sphere given by (1) to be conformal, the condition that the local linear scale factors along the meridian and along the parallel are equal must be met [8,17], i.e.,

$$-\frac{d\rho}{d\varphi} = \frac{m\rho}{\cos \varphi}. \tag{36}$$

Expression (36) can be written in the form of a differential equation

$$\frac{d\rho}{\rho} = -\frac{m d\varphi}{\cos \varphi}. \tag{37}$$

Integrating Equation (37) gives

$$\rho = K \tan^m\left(\frac{\pi}{4} - \frac{\varphi}{2}\right), \tag{38}$$

where $K > 0$ is a constant. Therefore, in the polar coordinate system, the equations of conic conformal projections read:

$$\theta = m\lambda, \ \rho = K \tan^m\left(\frac{\pi}{4} - \frac{\varphi}{2}\right). \tag{39}$$

Taking into account (10), we have

$$\rho_0 = K \tan^m\left(\frac{\pi}{4} - \frac{\varphi_0}{2}\right) = \frac{\cos\varphi_0}{m}, \tag{40}$$

and from there

$$K = \frac{\cos\varphi_0}{m} \cot^m\left(\frac{\pi}{4} - \frac{\varphi_0}{2}\right). \tag{41}$$

Furthermore,

$$x = \lim_{m\to 0}\rho(\varphi)m\lambda = \lim_{m\to 0}\frac{\tan^m\left(\frac{\pi}{4} - \frac{\varphi}{2}\right)}{\tan^m\left(\frac{\pi}{4} - \frac{\varphi_0}{2}\right)}\lambda\cos\varphi_0 = \lambda\cos\varphi_0, \tag{42}$$

$$y = \lim_{m\to 0}[\rho_0 - \rho(\varphi)] = \lim_{m\to 0}K\left[\tan^m\left(\frac{\pi}{4} - \frac{\varphi_0}{2}\right) - \tan^m\left(\frac{\pi}{4} - \frac{\varphi}{2}\right)\right] =$$

$$= \lim_{m\to 0}\frac{\cos\varphi_0}{m}\left[1 - \cot^m\left(\frac{\pi}{4} - \frac{\varphi_0}{2}\right)\tan^m\left(\frac{\pi}{4} - \frac{\varphi}{2}\right)\right] =,$$

with the application of L'Hôpital's rule

$$= \lim_{m\to 0}\cos\varphi_0\left\{\frac{-\cot^m\left(\frac{\pi}{4} - \frac{\varphi_0}{2}\right)\tan^m\left(\frac{\pi}{4} - \frac{\varphi}{2}\right)\ln\left[\cot\left(\frac{\pi}{4} - \frac{\varphi_0}{2}\right)\tan\left(\frac{\pi}{4} - \frac{\varphi}{2}\right)\right]}{1}\right\} =$$

$$= \cos\varphi_0 \ln\frac{\tan\left(\frac{\pi}{4} + \frac{\varphi}{2}\right)}{\tan\left(\frac{\pi}{4} + \frac{\varphi_0}{2}\right)} \tag{43}$$

So, the equations of the normal aspect cylindrical conformal projection, which is the limiting case of the normal aspect conical conformal projection of the unit sphere, are

$$x = \lambda \ \cos\varphi_0, \ y = \cos\varphi_0 \ln\frac{\tan\left(\frac{\pi}{4} + \frac{\varphi}{2}\right)}{\tan\left(\frac{\pi}{4} + \frac{\varphi_0}{2}\right)}. \tag{44}$$

If we translate the image of the projection by the amount $\cos\varphi_0 \ln\tan\left(\frac{\pi}{4} + \frac{\varphi_0}{2}\right)$ in the direction of the $y$ axis, we will achieve that the image of the equator is on the coordinate axis $x$ as is usual in the cartographic literature. Therefore, the final equations of the normal aspect cylindrical conformal projection, which is the limiting case of the normal aspect conical conformal projection, are

$$x = \cos\varphi_0 \cdot \lambda, \ y = \cos\varphi_0 \ln\tan\left(\frac{\pi}{4} + \frac{\varphi}{2}\right). \tag{45}$$

According to (12), for the projection (45) we have

$$h(\varphi) = k(\varphi) = \frac{\cos\varphi_0}{\cos\varphi}. \tag{46}$$

So, it really is a normal aspect cylindrical conformal projection.

*Example 3*

Let the conic projection be given by the equations

$$\theta = m\lambda, \ \rho = \frac{1}{2m\tan^m\left(\frac{\pi}{12}\right)}\tan^m\left(\frac{\pi}{4} - \frac{\varphi}{2}\right), \tag{47}$$

where $m$ is a real number, $0 < m < 1$. The length of the parallel corresponding to the latitude $\frac{\pi}{3}$ will be on the unit sphere equal to $\cos\frac{\pi}{3}\cdot 2\pi = \pi$, and defined by (47) in the projection plane $\rho\left(\frac{\pi}{3}\right)m\cdot 2\pi = \frac{1}{2m}m\cdot 2\pi = \pi$.

The corresponding cylindrical projection will have equations according to (45)

$$x = \frac{\lambda}{2}, \; y = \frac{1}{2} \ln \tan\left(\frac{\pi}{4} + \frac{\varphi}{2}\right), \tag{48}$$

and the length of any parallel in that projection (because they are all the same length) will be $2x(\pi) = 2\frac{\pi}{2} = \pi$. Thus, the cylindrical projection (48) is the limiting case of the conic projection (47) when $m \to 0$, and both map the parallel of latitude $\frac{\pi}{3}$ so that its length in the plane of both projections is as long as that parallel on the unit sphere, i.e., $\pi$.

## 6. Gnomonic Perspective Conic Projection

The equations of the gnomonic perspective conical projection in the polar coordinate system can be written in the following form:

$$\theta = m\lambda, \; \rho = K[\cot\alpha - \tan(\varphi - \alpha)], \tag{49}$$

where $K > 0$ is a constant, and $\sin\alpha = m$.

Considering (10), we have

$$\rho_0 = K[\cot\alpha - \tan(\varphi_0 - \alpha)] = \frac{\cos\varphi_0}{m}, \tag{50}$$

and from there

$$K = \frac{\cos\varphi_0}{m[\cot\alpha - \tan(\varphi_0 - \alpha)]}. \tag{51}$$

Furthermore,

$$x = \lim_{m \to 0} \rho(\varphi) m\lambda = \lim_{\alpha \to 0} \frac{\cot\alpha - \tan(\varphi - \alpha)}{\cot\alpha - \tan(\varphi_0 - \alpha)} \lambda \cos\varphi_0 = \lambda \cos\varphi_0, \tag{52}$$

$$y = \lim_{m \to 0} [\rho_0 - \rho(\varphi)] = \lim_{\alpha \to 0} K[\tan(\varphi - \alpha) - \tan(\varphi_0 - \alpha)] =$$

$$= \lim_{\alpha \to 0} \frac{\cos\varphi_0}{\sin\alpha} \frac{\tan(\varphi - \alpha) - \tan(\varphi_0 - \alpha)}{\cot\alpha - \tan(\varphi_0 - \alpha)} = \cos\varphi_0(\tan\varphi - \tan\varphi_0). \tag{53}$$

Therefore, the equations of the normal aspect cylindrical gnomonic perspective projection, which is the limiting case of the normal aspect gnomonic perspective conical projection of the unit sphere, read:

$$x = \lambda \; \cos\varphi_0, \; y = \cos\varphi_0(\tan\varphi - \tan\varphi_0). \tag{54}$$

If we translate the image of the projection by the amount $\sin\varphi_0$ in the direction of the $y$ axis, we will achieve that the image of the equator is on the coordinate axis $x$ as is usual in the cartographic literature. Therefore, the final equations of the normal aspect cylindrical gnomonic perspective projection, which is the limiting case of the normal aspect conic gnomonic perspective projection, are

$$x = \cos\varphi_0 \cdot \lambda, \; y = \cos\varphi_0 \tan\varphi. \tag{55}$$

According to (12) for the projection (55), we have

$$h(\varphi) = \frac{\cos\varphi_0}{\cos^2\varphi}, \; k(\varphi) = \frac{\cos\varphi_0}{\cos\varphi} \tag{56}$$

*Example 4*

Let a conic projection equation be given by

$$\theta = m\lambda, \; \rho = \frac{1}{2m} \frac{\cot\alpha - \tan(\varphi - \alpha)}{\cot\alpha - \tan\left(\frac{\pi}{3} - \alpha\right)}, \tag{57}$$

where $m = \sin \alpha$ is a real number, $0 < m < 1$. The length of the parallel corresponding to the latitude $\frac{\pi}{3}$ will be on the unit sphere equal to $\cos \frac{\pi}{3} \cdot 2\pi = \pi$, and defined by (57) in the projection plane $\rho\left(\frac{\pi}{3}\right) m \cdot 2\pi = \frac{1}{2m} m \cdot 2\pi = \pi$. The corresponding cylindrical projection will have equations according to (55)

$$x = \frac{\lambda}{2}, \ y = \frac{1}{2} \tan \varphi, \tag{58}$$

and the length of any parallel in that projection (because they are all the same length) will be $2x(\pi) = 2\frac{\pi}{2} = \pi$.

Therefore, the cylindrical projection (58) is the limiting case of the conic projection (57) when $m \to 0$, and both map the parallel of latitude $\frac{\pi}{3}$ so that its length in the plane of both projections is as long as that parallel on the unit sphere, i.e., $\pi$.

## 7. Projections Equidistant along Parallels

For the normal aspect conical projection of the unit sphere given by (1) to be equidistant along the parallels, the following condition should be satisfied:

$$k(\varphi) = \frac{m\rho}{\cos \varphi} = 1. \tag{59}$$

It follows directly from Equation (59) that

$$\rho = \frac{\cos \varphi}{m}. \tag{60}$$

So, in the polar coordinate system, the equations of the conic projection that is equidistant along the parallels for a sphere of radius 1 read:

$$\theta = m\lambda, \ \rho = \frac{\cos \varphi}{m}. \tag{61}$$

Consider

$$x = \lim_{m \to 0} \rho(\varphi) m\lambda = \lambda \cos \varphi. \tag{62}$$

It follows from (62) that $x$ is not a function of $\lambda$ alone, so there is no normal aspect cylindrical projection equidistant along the parallels that would be obtained as a limiting case of normal aspect conical projection equidistant along the parallels.

## 8. Conclusions

Lapaine [15] explained the shortcomings of the existing derivations of the equations of cylindrical projections which are limiting cases of conic projections. Then he interpreted and supplemented Lambert's derivation, which leads from a conic conformal projection to a cylindrical conformal projection, i.e., the Mercator projection. Following Lambert's idea, he showed that not only cylindrical conformal projections, but also cylindrical equivalent projections and cylindrical projections equidistant along meridians can be derived as limiting cases of corresponding conic projections. Lapaine [16] addresses the same problem, but this time supplements the explanations and extracts of Hinks [12]. In these derivations, Lapaine assumes that an equidistantly mapped parallel in the conic projection will be mapped to an equidistantly mapped parallel in the corresponding cylindrical projection.

In regard to the interpretation of cylindrical and azimuthal projections as limiting cases of conic projections, auxiliary developable surfaces are usually used. If the angle at the top of the cone becomes smaller and equals zero at one point, the cone moves in a straight line, but if at the same time one of its cross-sections maintains its size, the limiting case is a cylindrical surface. The conclusion is that if conic projections are mappings to a cone, then the limiting case of those projections should be cylindrical projections. In this paper, it is shown that this is not always the case, so it is another proof of the claim that the use of intermediate surfaces in the theory of cartographic projections is generally not

recommended. Therefore, in previous papers [15,16] and in this article, the author does not use development surfaces, except when the projection is really defined as a mapping to an auxiliary surface.

In previous papers that dealt with cylindrical projections as limiting cases of conic projections, standard or equidistant mapped parallels were used in the derivations. In this paper, it was shown that this approach is not necessary and that it is sufficient to use parallels that preserve length.

In the end, the fact was confirmed that the normal aspect cylindrical projection equidistant along the parallels cannot be obtained as a limiting case of the normal aspect conical projection equidistant along the parallels. This is obvious because a normal aspect cylindrical projection equidistant along the parallels does not exist. The limiting cases of conic projections are derived in this paper in the most natural way by deriving the equations of cylindrical projections from the equation of conic projections in a rectangular system in the projection plane using a mathematical concept of limits, and it is shown that such an approach is possible, but not always, so that it should be used carefully or, even better, avoided in teaching and studying map projections.

**Funding:** This research received no external funding.

**Informed Consent Statement:** Not applicable.

**Data Availability Statement:** Data are contained within the article.

**Conflicts of Interest:** The author declares no conflict of interest.

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
