# Peer review of "Connection of Conic and Cylindrical Map Projections"

_ijgi, doi:10.3390/ijgi13040113_

Round 1
Reviewer 1 Report
Comments and Suggestions for Authors
In this paper, cylindrical map projections are presented as limiting cases of conical projection based on parallels that preserve length. The limiting cases of conic projections are derived by deriving the equations of cylindrical projections from the equation of conic projections in a rectangular system in the projection plane using the mathematical concept of limits. This paper is well structured and very clear in explaining and deriving the equations.
Specific remarks
line 203 page 6: ccos ->cos
line 242 page 8: please check the calculations in deriving that ρ(π/3) = 1/2m
Reviewer 2 Report
Comments and Suggestions for Authors
The logical flow of the article is clear and understandable (although I note that part of it has already appeared in the author's article cited as reference 15). I have two request/suggestion regarding the paper.
First, in a few years, another author will refer to this article in the introduction of his own work: as "Lapaine (2024) states: XXX". In place of XXX comes a sentence: this is a one-sentence summary of the present work. It is this one sentence that I would like to see (1) compiled and (2) placed in both in the abstract and the conclusion. If it is there, the major part of the conclusion (first three paragraphs) can go smoothly into the discussion chapter, and at the end I need this "XXX" sentence.
Second. The mandatory elements and structure of the abstract are the answers to the following five questions: (1) what did I do? (2) how did I do it? (3) where did I do it? (in this case: is there a geographical limit to the derivation used?) (4) what did it come out as? - this is the above-mentioned sentence "XXX", and (5) why is this important?
Reviewer 3 Report
Comments and Suggestions for Authors
The abstract is puzzling, especially “parallels that preserve length”—because scale varies with latitude, “preserve length” is confusing. And “Lambert does not perceive conic and cylindrical projections as projections on the surface of a cone or cylinder” is similarly problematic.
It is an odd way to start an article. I’m equally puzzled by the second paragraph of the article. At least you bring in the concept diagram, albeit a bit late. It really is part of the direct quotation attributed to Monmonier.
Comments on the Quality of English Language
ok
Round 2
Reviewer 3 Report
Comments and Suggestions for Authors
As I communicated to the editors, I don't think adequate changes were made to accept the article as is.